# Extraction of Bioactive Compounds from Different Vegetable Sprouts and Their Potential Role in the Formulation of Functional Foods against Various Disorders: A Literature-Based Review

**DOI:** 10.3390/molecules27217320

**Published:** 2022-10-28

**Authors:** Afifa Aziz, Sana Noreen, Waseem Khalid, Fizza Mubarik, Madiha khan Niazi, Hyrije Koraqi, Anwar Ali, Clara Mariana Gonçalves Lima, Wafa S. Alansari, Areej A. Eskandrani, Ghalia Shamlan, Ammar AL-Farga

**Affiliations:** 1Department of Food Science, Faculty of Life Sciences, Government College University, Faisalabad 38000, Pakistan; 2University Institute of Diet and Nutritional Sciences, Faculty of Allied Health Sciences, The University of Lahore, Lahore 54000, Pakistan; 3Faculty of Food Science and Biotechnology, UBT-Higher Education Institution, St. Rexhep Krasniqi No. 56, 10000 Pristina, Kosovo; 4Department of Epidemiology and Health Statistics, Xiangya School of Public Health, Central South University, Changsha 410017, China; 5Department of Food Science, Federal University of Lavras, Lavras 37203-202, Brazil; 6Biochemistry Department, Faculty of Science, University of Jeddah, Jeddah 21577, Saudi Arabia; 7Chemistry Department, Faculty of Science, Taibah University, Medina 30002, Saudi Arabia; 8Department of Food Science and Nutrition, College of Food and Agriculture Sciences, King Saud University, Riyadh 11362, Saudi Arabia

**Keywords:** sprout, bioactive compound, new extraction, functional product, pharmacological role

## Abstract

In this review, we discuss the advantages of vegetable sprouts in the development of food products as well as their beneficial effects on a variety of disorders. Sprouts are obtained from different types of plants and seeds and various types of leafy, root, and shoot vegetables. Vegetable sprouts are enriched in bioactive compounds, including polyphenols, antioxidants, and vitamins. Currently, different conventional methods and advanced technologies are used to extract bioactive compounds from vegetable sprouts. Due to some issues in traditional methods, increasingly, the trend is to use recent technologies because the results are better. Applications of phytonutrients extracted from sprouts are finding increased utility for food processing and shelf-life enhancement. Vegetable sprouts are being used in the preparation of different functional food products such as juices, bread, and biscuits. Previous research has shown that vegetable sprouts can help to fight a variety of chronic diseases such as cancer and diabetes. Furthermore, in the future, more research is needed that explores the extraordinary ways in which vegetable sprouts can be incorporated into green-food processing and preservation for the purpose of enhancing shelf-life and the formation of functional meat products and substitutes.

## 1. Introduction

Sprouts are germinated from seeds of crops such as radish, cereals (rice and legumes), soybeans, and trees (*Toonasinensis* and pepper). Sprouts have been a popular dish in China for over 5000 years and have now spread to other Eastern countries. Sprout consumption has increased in Western cultures due to a shift in lifestyle towards convenience and health [1]. Sprouts have become more popular worldwide because of their nutritional value and health advantages. As compared with adult edible plant portions, sprouts are abundant in health-promoting bioactive chemicals, vitamins, and minerals. Sprouting is a food processing method that boosts the nutritional value of cereals, oilseeds, and vegetable seeds [2], by inducing macronutrient breakdown and increasing the amounts of amino acids, simple sugars, and other nutritional components [3]. Sprouting also helps to reduce anti-nutritional components and to improve sprouts’ digestibility and sensory aspects. Sprouts aid in the synthesis of new beneficial components such as polyphenols and vitamin C. Recently, sprouts have grasped consumers’ consumption interests due to their functional properties and phytonutritional profile. Sprouts have gained popularity as a top healthy food [4]. Due to the presence of biologically active compounds [5], sprouts have an important role in preventing several forms of malignancies. In the literature, sprouts have been shown to have a substantial anti-genotoxic impact against DNA damage [6]. According to Gawlik-Dziki et al. [7], Brassica and vegetable sprouts help to minimize the incidence of lung and colorectal cancer occurrence. According to epidemiological studies, the consumption of broccoli sprout-rich foods has been linked to a lower incidence of many malignancies and chronic degenerative diseases [8]. Cowpea sprouts have been celebrated for reducing cell proliferation and boosting anti-colorectal cancer activity [9]. Isoflavonoids in soybean sprouts defend against cancer and cardiovascular disease [10]. Sprouts include a variety of nutrients that are beneficial to human health and help to avoid a variety of diseases [11]. According to the research, sprouts are an excellent source of a range of phenolic compounds that protect against oxidative reactions. Mung bean sprouts have been reported to lower gastrointestinal issues and heart stroke [12]. Soybean sprouts have been demonstrated to have health-promoting qualities such as lowering cancer and cardiovascular disease risk [13]. Sprouts have been discovered to offer antidiabetic properties. In studies, *Brassica oleracea* sprouts have been shown to have antidiabetic, hepatoprotective, and antioxidant properties. The extracts had a lowering effect on blood glucose levels in the body and hepatoprotective and antioxidant properties. As a consequence, Brassica sprouts have anti-hyperglycemic activity [14]. Antibacterial activity has also been discovered in sprouts. Broccoli and pea sprouts have antibacterial properties against *Helicobacter pylori* (bacteria linked to stomach cancer) [15,16]. The purpose of this article is to explore methods for extracting bioactive components from certain vegetable sprouts to make functional meals and their bioavailability against certain diseases.

## 2. Bioactive Components in Different Vegetables Sprouts

Sprouts are recognized due to their high concentration of bioactive compounds [17]. Bioactive chemicals are present in high concentrations in food and food waste. Several bioactive compounds have been found with a wide range of functional and structural features [18] (Figure 1).

Natural chemical components present in minute amounts in plants are called bioactive chemicals [19]. These substances can interact with one or more live tissue components, resulting in many conceivable outcomes. Bioactive chemicals are naturally occurring essential and nonessential substances present in the food chain that have been shown to impact human health. Antioxidants and polyphenols have been studied for their many biological actions. Due to their unique properties, polyphenols such as protocatechuic acid, vanillic acid, caffeic acid, quercetin, and kaempferol have been shown to protect against oxidative damage [20]. Polyphenols are abundant in sprouts [21]. Polyphenols are reducing agents, hydrogen-donating antioxidants, and singlet oxygen quenchers formed in response to biotic or abiotic stress. Polyphenols’ multifunctionality is due to their dispersion in various tissues and organs of plants at various concentrations. Plants produce phenolic substances that contain at least one aromatic ring with one or more hydroxyl substituents and can be divided into flavonoids and phenolic acids based on their chemical structure. In recent years, phenolic compounds have been extensively studied for their antioxidant, anthelmintic, antiallergenic, anticancer, anti-inflammatory, antiviral, antiulcer, anti-hepatotoxic, antidiarrheal, and antiproliferative properties [3,22]. Micronutrients (vitamins and minerals) are important for tissue maintenance, bone and tooth production, and general health. They help to regulate and coordinate most biological activities and other biochemical and physiological functions by acting as cofactors and coenzymes in diverse enzyme systems. Humans and other creatures require micronutrients at varying levels throughout their lives to coordinate numerous physiological activities and sustain health [23,24]. Vitamins are essential for optimal health and perform vital functions in the human body. Vitamin C, often known as ascorbate, is a micronutrient that humans require. In humans, vitamin C deficiency inhibits the function of several enzymes and can lead to scurvy. Ascorbic acid is a cofactor in a variety of essential enzymatic processes [25], and it is involved in collagen formation. Vitamin E (tocopherols and tocotrienols) protects DNA, low-density lipoproteins, and polyunsaturated fatty acids against oxidative damage. Vitamin E is also involved in hemoglobin production, immune response regulation, and membrane structure stability, and it helps to keep blood coagulation, bone growth, and healing under control. In newborns, vitamin K deficiency can cause hemorrhagic illness and surgical bleeding, muscular hematomas, and intracranial hemorrhages in adults [26].

Minerals are required to perform processes that are necessary for a healthy life. The human body needs calcium for optimal heart and muscle function, bone production, and blood cell creation and function. Copper, molybdenum, selenium, and zinc are key components of various critical enzymes in the human body. In contrast, iron is required for several protein syntheses, including hemoglobin, which helps to avoid anemia. Magnesium is necessary for ATP processing and bone health. Sodium and potassium are electrolytes found throughout the body and are necessary for the coregulation of ATP. Phosphorus is found in bones and cells, and it also plays a role in energy metabolism, DNA and ATP (as phosphate), and a variety of other functions [27]. The extraction of bioactive compounds from different types of vegetable sprouts is shown in Table 1.

## 3. Conventional and New Extraction Methods Are Used to Extract the Bioactive Compounds

Several traditional extraction procedures can extract bioactive chemicals from plant sources. The majority of these methods rely on a solvent’s ability to extract and the use of heat and/or mixing. The three most common procedures for extracting bioactive chemicals from plants are Soxhlet extraction, maceration, and hydro distillation [43]. There are both traditional and modern ways of obtaining isoflavones from plants. Maceration, percolation, decoction, infusion, Soxhlet extraction, and hot reflux extraction are all examples of traditional extraction procedures that have been employed to extract bioactive chemicals [44]. As a result, developing quick, safe, and environmentally acceptable technology for analyzing and separating bioactive chemicals is critical. Isoflavones have been separated via “ultrasound-assisted extraction (UAE), microwave-assisted extraction (MAE), supercritical fluid extraction (SFE), and pressured liquid extraction with green solvents such as ionic water liquids and supercritical carbon dioxide (PLE)”. These methods use organic solvents, take less time, and produce higher yields and quality [45].

In recent years, natural deep eutectic solvents (DESs) have received much attention as good green solvents for extracting bioactive chemicals from natural resources [46]. The current research has looked at the feasibility and effectiveness of extracting isoflavones from chickpea sprouts using various polarities of natural deep eutectic solvents (DESs), by testing 20 different DESs that included hydrogen bond acceptors such as “choline chloride, betaine, and L-proline with different hydrogen bond donors (carboxylic acids, alcohols, sugars, and amine). The researchers looked at the yields of four isoflavones (ononin, sissotrin, formononetin, and biochanin A), total flavonoid concentration, and antioxidant activity to estimate extraction efficiency. Using a Box–Behnken design in conjunction with response surface techniques, “the components that contribute to optimal ultrasound-assisted extraction conditions were then examined.” The extraction yields of isoflavones were significantly affected by DES water content and extraction temperature. Our findings suggest that DESs might be utilized to extract bioactive chemicals from a variety of biomaterials [47]. Sprouts from peanuts yield trans-resveratrol in accelerated solvent extraction [48].

Soxhlet extraction has been used to recover a large number of phytochemicals from *Azadirachta indica* (Neem) leaf powder, predominantly nonpolar components [49]. Evaluation of Soxhlet extraction for *Moringa oliefera* leaves resulted in lower yield, as well as phenolic and flavonoid contents [50]. *Centella asiatica* extraction was optimized using Soxhlet extraction, which produced the best results at 25 °C, a sample-solvent ratio of 1:45, 200 rpm agitation speed, and 1.5 h [51]. After removing lipoidal components from powdered *Clitorea ternate* flowers with petroleum ether at 60–80 °C, the yield was 2.2 percent *w*/*w* [52].

After more ethanol extraction from the marc, alkaloids and saponins were confirmed to be present. However, the anthocyanin, i.e., the main pigment of *Clitorea ternate* flowers, was not present, indicating that oxidation and degradation had taken place. As compared with other solvents such as petroleum ether, chloroform, and water, the extraction of *Psidium guajava* L. [53] leaves using ethanolic and hydro alcohol extracts (4:1 *v*/*v*) produced the highest extraction yield with the greatest presence of phytoconstituents (alkaloids, saponins, carbohydrates, tannins, and flavonoids) [50]. Nonpolar solvents such as petroleum ether and chloroform revealed no retained active chemicals and very low tannin content in the extracts, respectively. With the exception of the absence of any alkaloids, water was shown to be as effective as ethanol. Polar solvents have been shown to work better for removing bioactive compounds from *Psidium guajava* [54]. As compared with aqueous extracts of *Garnicia atriviridis*, methanol extracts (1:10 *w*/*v*) had stronger antioxidant activities, while the aqueous extracts had better anti-hyperlipidemic activities [50]. Based on total phenolics, maceration with various solvents at a ratio of 1:10 *w*/*v* sample to solvent, for an hour, revealed that 70% acetone was an effective solvent for Portucala oleracea and 70% methanol was an effective solvent for flavonoids in *Cosmos caudatus* [55]. As compared with Soxhlet extraction and percolation using a comparable solvent, maceration with 70% ethanol and powdered dried materials at 1:40 *w*/*v* showed the greatest phenolic and flavonoid concentrations for *Moringa oliefera* [56]. Using 100% ethanol as the solvent at 75 °C and an irradiation power of 600 W for four cycles, MAE has been evaluated as a new technique to extract triterpene from *Centella asiatica* and the yield was increased by two times over Soxhlet extraction [50]. Combining MAE with enzyme lysis (such as cellulase) has been shown to increase extraction; the ideal conditions of sample/solvent ratio at 1:36, enzyme pretreatment at 45 °C for 30 min, and irradiation at 650 W for 110 s produced a yield of 27.10 percent. However, Trusheva et al.’s observation that an extra MAE cycle had an impact on the phytochemical degradation was not examined. The best extraction was obtained using MAE with 100 W and 1:12.5 sample/solvent ratios on *Dioscorea hispida*. The UAE has been shown to be the most productive technique for extracting propolis based on its high yield, lengthy (10–30 min) extraction duration, and excellent selectivity. To shorten extraction times and to prevent exposure to high temperatures, UAE was used to extract thermolabile chemicals such as anthocyanin from floral components. When extracting *Withania somnifera* using water as the solvent for 15 min, the yield was at its highest, reaching 11.85 percent as compared with ethanol and water-ethanol at various 5, 15, and 20 min extraction times. A higher effectiveness on phenolics was observed when *Cratoxylum formosum* was extracted using ultrasonic at 45 kHz, 50.33 percent ethanol by volume, at 65 °C for 15 min. Free radical production at irradiation frequencies higher than 20 kHz, however, may need to be taken into account [50].

Sprouts and microgreens are edible seedlings of different vegetables and herbs that have become increasingly popular due to their positive health benefits and are now referred to as functional foods or superfoods. Bioactive components have long been appreciated in broccoli seedlings (*Brassica oleracea* L. var. *Italica*). Secondary metabolites have been linked to several positive health outcomes. In in vitro and animal studies, broccoli seedlings have been shown to have health benefits. A current study has summarized previous research on the bioactive components and bioactivities of various broccoli derivatives, as well as the mechanisms of action associated with them [57]. Conventional and new techniques can be used to extract bioactive compounds from different types of vegetable sprouts, as shown in Figure 2.

## 4. Food Applications of Vegetable Sprouts

Nowadays, the food industry has been focused on developing healthier products that are more responsive to changing customer demands. Recently, sprouted grains have become a new element in the culinary world. Sprouted grains have a higher nutritional value, lower antinutrient content, a prime source of bioactive compounds, and a sweeter flavor, making them a potential new food component [58]. Sprouted grains were formerly only used in bread, but they may now be found in tortillas, granola, cookies, crackers, muffins, snacks, bars, morning cereals, side dishes, and salads [59]. Sprouted grains may be used in various culinary applications without requiring any formulation adjustments, and they can assist considerably in differentiating products.

After sprouting and drying, a whole grain kernel can be milled into flour or processed into grits, coarse meals, or flakes, among other granulations. Wheat, rye, spelt, barley, brown rice, oat, sorghum, millet, quinoa, buckwheat, and amaranth may all be sprouted and used in several nutritious applications as long as the germ is intact. Bars, cereals, granola, bread, tortillas, frozen dough, candies, snacks, side dishes, soups, and pasta are common components. Gluten-free foods such as sprouted sorghum, millet, quinoa, amaranth, buckwheat, brown rice, and purity protocol oats are naturally gluten-free foods. They can be added to gluten-free diets to boost nutrition. Because of the wide variety of grains available, bakers, food scientists, and chefs have a lot of creative freedom. The functional differences between sprouted grains and their unsprouted counterparts must be recognized and addressed for optimal formulation, processing, and end-product attributes. Due to their high nutritional content, interesting technical possibilities, and sensory qualities, sprouted grains are being exploited as a component in a variety of food product innovations. The quality of sprouts and their specific culinary behavior depends on the germination circumstances. In this article, we look at two applications of sprouted grains: the effect of wheat sprouting time on the production of innovative baking flours and the microbiological risk of homemade rejuvelac, a sprouted wheat-based fermented beverage [60]. Bakers may notice a shortening of the proofing period or an increase in bread absorption, contributing to higher yields.

In tortillas, flour made of sprouted whole wheat can help to soften them, lengthen their shelf-life, and improve their sensory qualities [40]. The potential of sprouted wheat to improve the likability of bread and tortillas might lead to an increase in whole grain consumption, which would be a tremendous step forward in human health, especially because these staples are frequently consumed on a regular basis [52]. Table 2 shows the food applications of vegetable sprouts.

## 5. Bioavailability of Sprout against Different Diseases

With the intake of plant sprouts, bioavailability has long been regarded as crucial. The bioavailability of phytochemicals in various sprout diets varies substantially depending on several parameters. Interindividual factors, including delivery mode, and even intraindividual biochemical variances and the makeup and function of the gut microbiota are all factors to consider. In one study, to test iso-thiocyanate bioavailability, mice were administered either thermally processed broccoli sprout powders or pure isothiocyanate sulforaphane. The greatest quantities of the isothiocyanate metabolite were discovered in slightly cooked broccoli sprout powdered meals. In vivo, nonheated broccoli sprouts were followed by powdered broccoli sprout meals. They identified erusin and sulforaphane interconversion and observed that erusin was the preferred form in the kidney, liver, and bladder even when just sulforaphane was digested. It is worth mentioning that the bioavailability of sulforaphane from broccoli sprouts varies greatly depending on the delivery method [68]. The study suggested the inhibitory effect of broccoli sprout extracts on the properties of two prostate cancer cell lines characterized by low (AT-2) and high (MAT-LyLu) metastatic potential. These effects may be due to the fact that broccoli sprouts contained flavonoids and phenolic acids [7]. A previous study suggested that fava bean sprouts had higher antioxidant activity because they contained more polyphenols and l-3,4-dihydroxyphenylalanine (l-DOPA) than the bean itself [69]. The study suggested that lentil sprouts contained melatonin that is a multifunctional antioxidant neurohormone. The results showed that germination of lentils increased the content of melatonin. In another study, Sprague Dawley rats were used to investigate the pharmacokinetic profile of melatonin after oral administration of a lentil sprout extract and to evaluate plasma and urine melatonin and related biomarkers and antioxidant capacity. The outcomes showed that lentil sprout intake increased melatonin plasmatic concentration and attenuate plasmatic oxidative stress [70]. Figure 3 shows the bioavailability of sprouts against different disorders.

### 5.1. Bioavailability of Sprouts against Brain Issues

Nervous system diseases are a common ailment that will become more common as populations age. Axonopathy, also known as dying-back axonopathy, is a neurological illness in which axons become disconnected from their destinations, resulting in functional impairment. Axons can renew or sprout in response to several neurologic illnesses to re-establish synaptic function and to reconnect with the target before motor neuron death. Compensatory motor axon sprouting and neuromuscular junction reinnervation has been demonstrated in ALS patients, although the disease’s course has typically outpaced these advantages. In ALS and kindred illnesses defined by dying-back axonopathy, potential therapeutics that encourage compensatory sprouting and reinnervation may delay symptom onset and may sustain muscle function for extended periods. Many questions concerning the impact of various disease-causing mutations on axonal outgrowth and regeneration, especially in motor neurons derived from patient-induced pluripotent stem cells, remain unsolved. Researchers must mimic the human neuromuscular circuit using motor neurons created from human-induced pluripotent stem cells to uncover drugs that stimulate axonal regeneration, sprouting, and reinnervation of neuromuscular junctions [71]. Regarding colored flavonoids, anthocyanins, the majority of which are highly acylated, and glycosylated forms of cyanidin are abundant in broccoli, radishes, cabbages, and kale sprouts [72]. Recently, anthocyanins have attracted more attention due to their potential to improve brain function and their role in the prevention and treatment of disorders including diabetes and obesity. One study suggested that two crude juices of broccoli sprouts had a protective effect on SH-SY5Y cells treated with the fragment Aβ25–35 because they contained different amounts of polyphenols and sulforaphane. The sprouts’ juices both protected against Aβ-induced cytotoxicity and apoptotic cell death as evidenced by cell viability, nuclear chromatin condensation, and apoptotic body formation measurements [73]. Another study suggested that cruciferous vegetables were a good source of sulforaphane. The results of this study showed that sulforaphane protected against acute brain injuries and neurodegenerative diseases through activating the Nrf2 signaling pathway [74].

### 5.2. Compensatory Sprouting as a Potential Therapeutic Strategy for Amyotrophic Lateral Sclerosis

Functional motor recovery can be aided by the sprouting of motor axons and the reinnervation of denervated NMJs. Axonal sprouting allows motor units to increase 5–8 times their initial size. In amyotrophic lateral sclerosis, there is evidence of motor axon sprouting [75]. ALS is more common in certain motor neuron subpopulations that are also less prone to sprouting. In people with amyotrophic lateral sclerosis, compensatory sprouting may be employed to slow the onset of muscle denervation and weakness. The global number of ALS cases is expected to rise by 2040. Any drug that can improve the quality of life for ALS patients is badly needed. Axonal sprouting, which involves the functional reinnervation of NMJs, has the potential to improve life quality [76]. Phenolic acids are present in different vegetable sprouts. The diverse neuroprotective effects of phenolic acids make them interesting candidates for better ALS therapies. Study outcomes have shown that protocatechuic acid administration at 100 mg/kg in SOD1G93A mice prolonged survival, recovered motor functions, and decreased gliosis [4,77]. An in vitro study suggested that antioxidant molecules were capable of rescuing NSC34 motor neuron cells expressing an ALS-associated mutation of superoxide dismutase 1 [78].

### 5.3. Bioavailability of Sprouts against Gastrointestinal Tract (GIT) Health Problems

Sprouts may make it easier for you to digest your diet. According to a study, sprouted seeds increased the amount of fiber in them, making them more accessible. According to one study, cereals sprouted for five days had up to 133 percent more fiber than non-sprouted grains. Another study found that growing beans until the sprouts were 5 mm long boosted the overall fiber content by 226 percent. Sprouting appears to enhance the amount of insoluble fiber, a type of fiber that aids stool creation and passage through the stomach, reducing constipation risk. Finally, sprouted beans, grains, vegetables, nuts, and seeds have lower antinutrient levels than their non-sprouted counterparts. This makes it easier for the body to absorb nutrients during digestion [79]. Fiber that the human gut cannot digest on its own, but some bacteria can digest, is an essential source of nutrients that your gut microbe need to stay healthy. Fiber helps to stimulate the growth of colonic flora, to increase the weight of the stool, and to enhance the number of bacteria in the gut. The growth of bacteria present in the gut enhances the health of the intestines. However, short-chain fatty acids are produced by anaerobic gut bacteria through saccharolytic fermentation of complex resistant carbohydrates, which escape digestion and absorption in the small intestine [80]. In contrast to micro- and macronutritional contents, dietary polyphenols tend to be recognized as xenobiotic by humans during absorption, and therefore, their biological accessibility is significantly low. Furthermore, polymerization and structural complexity influence digestion in the small intestine [81]. The small intestine usually consumes approximately 5–10% of the absorbed polyphenols. The residual polyphenols (90–95%) might develop up to millimolar proportions in the large intestine linked to bile conjugates spilled into lumen in which they are susceptible to the enzymatic reactions of the gut bacteria species [82]. According to current data, dietary polyphenols that penetrate gut microflora, also including volatile compounds produced, manufacture and generate differences in the microbiota community through their prebiotic properties and functioning as an antiseptic towards infectious intestinal microbiota [83].

Onions have been proven to offer digestive system-protective properties, such as preventing stomach ulcers, regulating gut flora, and alleviating colitis. In rats, raw onion sprouts were shown to suppress histamine-induced stomach acid release and to attenuate ethanol-stimulated gastric ulcers. However, boiling the onion was less effective. In common carp juveniles, dietary supplementation with onion sprout powder has been shown to alter gut microbiota by increasing the number of lactic acid bacteria [84]. In rats, bioactive substances produced from onions, such as quercetin and quercetin monoglycosides, were found to boost the enzymatic activity of the gut microbiota. In colitis mice caused by dextran sodium sulfate, quercetin monoglycosides were shown to affect a variety of gut bacteria. Furthermore, onions and other *Allium* species have been demonstrated to protect against upper aerodigestive tract and gastrointestinal tract cancers [85]. Peanut sprout ethanolic extract at a purification of 80% (*v*/*v*) has been administered to loperamide-induced constipated SD rats, which revealed its laxative effects [86].

### 5.4. Bioavailability of Sprouts against Cardiovascular Diseases (CVDs)

In hypercholesterolemic Wistar rats, dietary supplementation with onion reversed high-cholesterol diet-induced changes in lipid mediators such as oxylipin and sphingolipid profiles [87]. Using an animal model to study increased blood pressure, researchers investigated the relationship between oxidative stress and a diet rich in broccoli sprouts with a high quantity of glucoraphanin. After 14 weeks, rats were fed broccoli sprouts that were either low in the chemical or rich in glucoraphanin. After the trial, they observed that rats fed a glucoraphanin-rich diet had lower blood pressure and less heart inflammation. According to the researchers, the benefits were attributed to better antioxidant defense systems and a decreased glucoraphanin-induced inflammatory response. Broccoli and broccoli sprouts contain different antioxidants (vitamin E, β-carotene, α-tocopherol, and ascorbic acid) that may aid in the prevention of cardiovascular diseases. In laboratory rats, the chemical glucoraphanin increased heart function, decreased inflammation, and boosted natural antioxidant defenses. When unstable molecules, called free radicals, react with oxygen in the body, they promote inflammation and cell death, raising the risk of heart disease and cancer. Antioxidants are supposed to help reduce oxidative stress in the body, preventing these detrimental consequences. Glucoraphanin is a chemical that boosts the body’s antioxidant defenses by acting as an indirect antioxidant. It is naturally found in broccoli and broccoli sprouts [88].

Onions have been shown in trials to enhance lipid profiles and to prevent platelet aggregation, lowering the risk of heart disease. Onions and their bioactive components have been widely researched for their hypocholesterolemia effects in rats fed high-cholesterol or high-fat diets. Onion sprouts successfully reduced total cholesterol, triglyceride, and low-density lipoprotein cholesterol levels in hyperlipidemic rats [89]. Polyphenol-rich onion extract alleviated hyperlipidemia in Sprague-Dawley rats’ livers by upregulating the low-density lipoprotein receptor (LDLR) and downregulating the 3-hydroxy-3-methylglutaryl (HMG)-CoA reductase (HMGCR). In addition, Lee et al. [90] found that quercetin-rich onion peel extract increased fecal cholesterol, reduced the atherogenic index, cardiac risk factor, and activation of LDLR and cholesterol 7-monooxygenase (CYP7A1) in high-cholesterol diet-fed mice, indicating that onion had a cholesterol-lowering effect via fecal excretion. They proved that fecal excretion of onions lowered cholesterol. When onions were added to a high-cholesterol diet supplied to rats, the bile acid levels in their stools changed. Dietary onion increased antioxidant enzyme activity and enhanced anti-inflammatory response and cardiovascular risk markers in rats fed a high-cholesterol diet [66]. An overview of vegetable sprouts’ bioavailability against different diseases is given in Table 3.

### 5.5. Bioavailability of Sprouts against Oxidative Stress-Related Diseases such as Cancer and Diabetes

In addition, secondary metabolites are abundantly present in sprouts, especially the glucosinolates (GLs), as in the case of the *Brassicaceae* family [102]. Gulcosinolates consist of an amino acid group and a thiohydroximate-O-sulfonate attached to the glucose unit [103]. Myrosinase acts to hydrolyse these GLs to thiocyanates and isothiocyanates [89] when the pH is between 6.0 and 7.0 [79], and then it yields anti-mutagenic activity, having a limiting effect on oxidative stress and playing a role in chemoprotection, especially in cancers and diabetes [104]. Glucoraphenin and glucobrassicin are the GLs excessively present in sprouted radish, which readily enhance antioxidant activity, and consequently decrease carcinogenesis in the body [105]. Kale sprouts do not have dehydroerucin but have a better GL profile as compared with sprouted radish due to gluconapoleiferin, glucoiberin, gluconasturtin, gluconapin, progoitrin, glucobrassicin, neoglucobrassicin, 4-hydroxyglucobrassicin, and sinigrin, which potentially reduce oxidative stress, and hence, decrease the risk of related diseases, i.e., diabetes, cancer, and heart diseases [94]. Taniguchi et al. [106] used Japanese radish sprouts in normal and streptozotin-induced diabetic mice to show the benefits of cruciferous sprouts on DM. It was shown that radish sprout consumption decreased plasma levels of fructosamine, glucose, and insulin, suggesting that the hypoglycemia brought on by radish sprout consumption may not be related to an increase in insulin synthesis but rather to enhanced sensitivity or an insulin-like action [107]. It depends on the ktype of sorghum, enzyme-inducing, and anti-proliferative capabilities. The most effective inducer of quinone oxidoreductase, a phase II detoxifying enzyme, has been shown to be an extract from black tea (non-tannin) that is abundant in 3-deoxyanthocyanins. Comparatively speaking, white sorghum extract has been shown to be a relatively potent inducer. Despite not inducing quinone oxidoreductase, tannin sorghum extracts have provided the most potent antiproliferative effects on human esophageal and colon cancer cells.

### 5.6. Bioavailability of Protein against Malnutrition

Sprouting enhances protein content, as evidenced from a study conducted by Devi et al. [95] on cowpea (lobia) by enhancing its bioavailability and digestibility. Sprouting is an interesting phenomenon that influences metabolic enzymes, especially proteinases, which increase the content of protein [96]. Sprouted chickpeas have more protein content than black gram. The sprouting process decreases the protease inhibitors and even enhances lipase activity, yielding increased content of fatty acids, and this also improves the digestibility of starches [97].

## 6. Sprout Vegetables as an Ingredient or Substitute for Meat Products

Dried sprouted food ingredients have been a trend for healthy/functional foods to live healthier, particularly by incorporating them into bread making flours or traditional beverages and juices [108]. With continued research on the influence of sprouted dietary feed on animals, it has revealed increased phytochemicals, particularly antioxidants, in the animals’ meat, as well as enhanced fatty acid content, particularly when sprouted alfalfa and flax were fed to rabbits [54]. Meat product consumption to achieve protein requirements has increased, and currently, it is difficult to rely on just one livestock source. With advancements in in vitro meat technology, tissue culturing engineers have started to develop lab-grown meat [109].

## 7. Conclusions

It is concluded that sprouts have been introduced as a new food for some years. Vegetables are basically important plant-based foods. Vegetable sprouts are composed of bioactive compounds, including phenolic compounds, antioxidants, etc. These bioactive compounds are extracted from sprouts by using different conventional methods and new techniques. Plant protein content improvement and enhanced protein bioavailability and digestibility can set up a better opportunity to research and develop plant-based meat protein substitutes. Vegetable sprouts are being used to develop functional foods and they also play an important role in maintaining the stability of food products. Furthermore, pharmaceutically, they aid in the defense of different types of chronic disorders.

## 8. Future Prospective and Recommendations

Vegetable sprouts’ potential involvement in the prevention and treatment of chronic diseases has to be investigated further. The high nutrient content of sprouts may provide extra lipid-lowering advantages. Sprouts are fiber-rich foods that are likely to provide a feeling of fullness. It is also crucial to remember that functional meals must be consumed often in order to offer their somewhat modest benefits. To check the probable positive effect of vegetable sprout foods on chronic diseases or risk factors related to lifestyle, a comprehensive scientific human study is required. The influence of the whole meal, which represents the synergistic effect between components, must be explored by conducting different studies on extracts and components. It is equally crucial to consider the makeup of the background diet because it might bias results and could create challenges in connecting the effects to the fitting dietary elements. The effect of vegetable sprout-based meals on chronic diseases or risk factors related to lifestyle must reflect the whole diet in order to apply the trial’s findings in practice. Unfortunately, in vegetable sprout research, because metabolic changes and their link that may affect biological activity in the body after consumption have not been taken into consideration, it is difficult to characterize the direct antioxidant impact of vegetable sprouts. It is necessary to determine the safety of ingesting the quantities of vegetable sprout extracts utilized in these studies by dietary consumption of foods containing vegetable sprouts. However, information gained from many types of experimental studies has contributed to a broader understanding of how the vegetable sprout food matrix may be advantageous. Keeping in mind the gap between protein supply and demand, more time is needed to research and develop sprouted vegetable protein-based products, which would be a better approach because this source would provide a better choice of protein accompanied by phytochemicals.

## Figures and Tables

**Figure 1 molecules-27-07320-f001:**
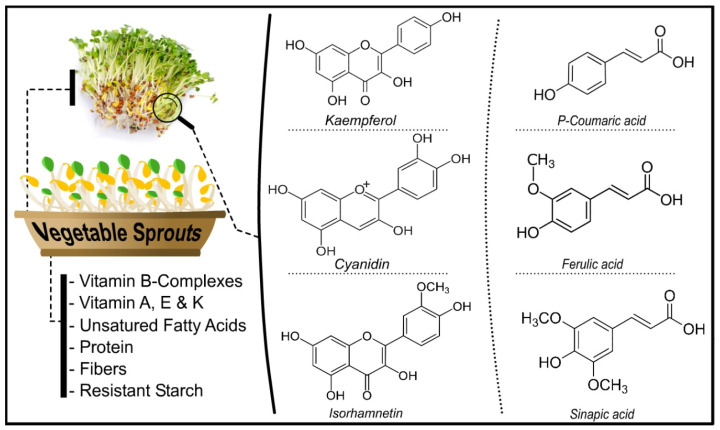
Different bioactive compounds in vegetables sprouts.

**Figure 2 molecules-27-07320-f002:**
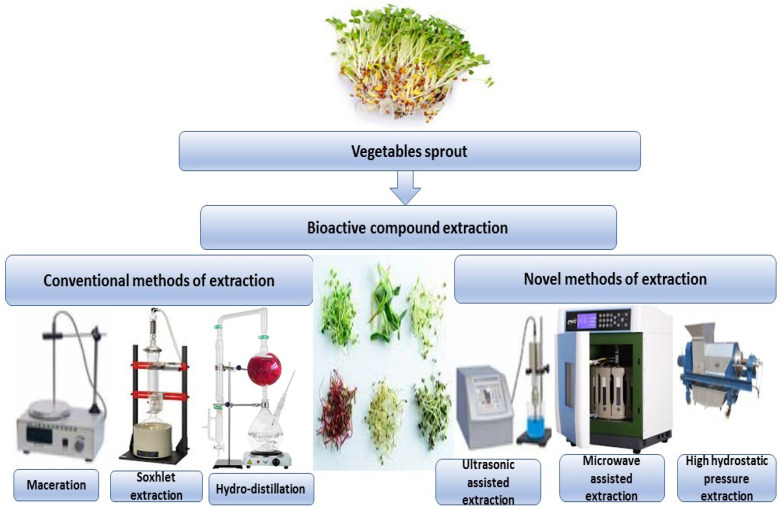
Conventional and new techniques can be used to extract bioactive compounds from different types of vegetable sprouts.

**Figure 3 molecules-27-07320-f003:**
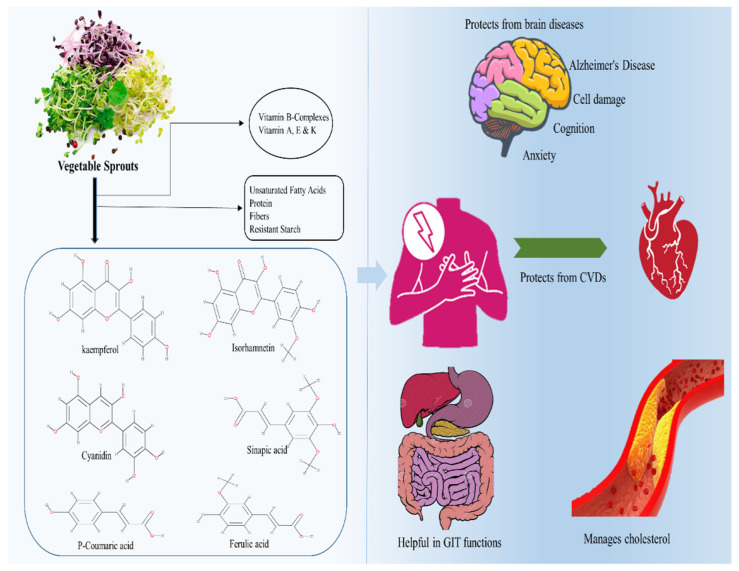
Bioavailability of sprouts against different disorders.

**Table 1 molecules-27-07320-t001:** Extraction of bioactive compound from vegetables sprouts.

Vegetable	Type	Extraction/Detection/Methods	Bioactive Compounds	Solvent	References
Water spinach	Leafy	--	Carotenoids and phenolic acids	--	[28]
Quinoa	Leafy	Conventional --	Phenolics, flavonoids, carotenoids (β-carotene and lycopene), and chlorophylls (a and b)	Ethanol	[29]
Brassica	Leafy	--	Polyphenols and glucosinolates	--	[30]
Kale and broccoli	Leafy	Ultrahigh-performance liquid chromatography high-resolution mass spectrometry	Polyphenols and glucosinolates	Methanol/water	[31]
Quinoa	Leafy	Spectrophotometry	Total phenolic compounds and antioxidants	Methanol	[32]
Radish, broccoli, leek, and beetroot	Leafy, Root	ABTS, FRAP, and ORAC	Polyphenols, L-ascorbic acid, carotenoids, and chlorophylls	Methanol	[33]
Garden cress	Leafy	HPLC, ABTS, and DPPH	Phenolic compound, antioxidant,, and flavonoids	Methanol	[34]
Onion	Root	HPLC and FTIR	Flavonoids, quercetin, and glucosides	Ethanol	[35]
Broccoli and red radish	Leafy Root	Conventional and HPLC-DAD	Glucosinolates and phenolic compounds	Methanol	[36]
Turnip	Root	Gas chromatography, mass spectroscopy, and DPPH	Phenolics, glucosinolates, and antioxidant,	Water	[37]
Fennel	Root	HPLC	Vitamin C, polyphenols, and antioxidants	Methanol/water	[38]
Brassicaceae	Root and leafy	FRAP	Phenolic compound and antioxidants	Methanol	[39]
Brussels	Leafy	HPLC and spectroscopic analysis	Chlorophyll, vitamin C, polyphenols, flavonoids, and antioxidants	Methanol	[36]
Sweet potato	Root	DPPH, spectrophotometer, LCMS/MS method	Anthocyanin and antioxidant,	Ethanol	[40]
Vegetable	Leafy	--	Glucosinolates, phenolics, and isoflavones	--	[41]
Brassica	Leafy	Conventional	Sulforaphane	--	[42]

**Table 2 molecules-27-07320-t002:** Food applications of vegetables sprouts.

Sprout Source	Application in Food Products	Improvement	References
Vegetable sprout	--	Improve nutritional value of different food product	[61]
Radish, red cabbage, vegetable green, buckwheat and broccoli seeds	--	Vegetable sprouts are rich in nutrients	[62]
Fresh alfalfa and flax sprouts	Rabbit meat	Modified the fat content, fatty acid, and phytochemical profile of the meat	[63]
Wheat seeds	Bread	To examined the profile of phenolic acids and antioxidant properties of wheat bread	[64]
Brown rice	Wheat bread	Sensory acceptance and longer shelf-life	[65]
Broccoli	Broccoli sprout juice	Broccoli sprouts are naturally enriched in glucoraphanin (GR)	[66]
Brussels	Juice	It may reduce the risk of cancer of the alimentary tract	[67]

**Table 3 molecules-27-07320-t003:** Bioavailability of sprouts against different diseases.

Vegetable Sprout Types	Study Design	Disease	Recovery	References
Cruciferous	Human (male and female)	Cancer	Cruciferous vegetables reduce the risk of cancer by decreasing the damage to DNA	[91]
Broccoli	--	--	The biological properties of broccoli are antioxidant, anticancer, anticancer, antimicrobial, anti-inflammatory, anti-obesity, and antidiabetic activities	[92]
Broccoli	Mice and rats	Alleviate pain	The broccoli sprouts have ability in pain therapy	[36]
Red cabbage, broccoli, Galega kale and Penca cabbage	,--	--	Different vegetables sprouts have antioxidant and anti-carcinogenic properties	[93]
Broccoli	Human	Cancer	Broccoli may reduce the risk of cancer by managing metabolism	[94]
Broccoli	Mice	Prostate tumorigenesis	Broccoli sprouts have significant inhibitory effects on prostate tumorigenesis.	[95]
Alfalfa	Mice	Inflammation	The study suggests that alfalfa supplementation can suppress the production of proinflammatory cytokines and alleviate acute inflammatory hazards.	[96]
Brussels	--	Cancer	Brussels sprouts have cancer preventive effects which may be due to a reduction in oxidative DNA damage	[97]
Spinach, kale, Brussels sprouts, mustard greens, green bell peppers, cabbage, and collards	Human	Binding of bile acids	The results show equal health-promoting potential of spinach, kale, brussels sprouts, mustard greens, green bell peppers, and collards, as indicated by their bile acid binding on dry matter basis	[98]
Brussels	Human	--	The results show that compounds in cooked and autolysed brussels sprouts can enhance lymphocyte resistance towards H2O2-induced DNA strand breaks in vitro	[99]
Radish, broccoli, leek, and beetroot	In vitro	Diabetic, obesity and cholinergic	Different vegetable sprouts can be used daily as superfoods or functional food	[33]
Turnip, cauliflower, and mustard	In vitro	Cancer	In vitro antiproliferative study supports that sprouts are a good source of anticancer agents	[100]
Broccoli	In vitro	Cancer	In vitro study indicates that broccoli sprouts can reduce prostate cancer	[101]

## Data Availability

Not available.

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
