# Peer review of "Extraction of Bioactive Compounds from Different Vegetable Sprouts and Their Potential Role in the Formulation of Functional Foods against Various Disorders: A Literature-Based Review"

_molecules, 2022, doi:10.3390/molecules27217320_

Round 1
Reviewer 1 Report
§ Line 30: "new technologies" modify to "advanced technologies," these approaches are not innovative.
§ Readers will find it fascinating and educational if a table is added to compare various extraction methods for the same sample (vegetable sprouts).
§ Recommendations and future and prospective planning are necessary.
§ Figures 1 and 3 express the same concept and may be combined.
§ Figs. 2 and 3: The resolution and quality have been improved.
§ Add the type of solvent used in Table 1.
§ In Tables 1, 2, and 3, replace "Authors" with "References"
§ Table 1: "Phytochemicals" is ambiguous; please clarify
§ Line 51: Replace ". [1]" with "[1].", then repeated for all references.
§ Line 412-415” Keeping in mind the gap between protein………. accompanied by phytochemicals.” Move to the conclusion section.
Author Response
Comments and Suggestions for Authors
- Line 30: "new technologies" modify to "advanced technologies," these approaches are not innovative.
Response: We appreciate the reviewer for the relevant information. Modified from new technologies to advanced technologies.
- Readers will find it fascinating and educational if a table is added to compare various extraction methods for the same sample (vegetable sprouts).
Response: We are thankful to the respectable reviewer for valuable comment. In extraction section a Table and Figure already added. We did read detail literature about vegetables sprout but the comparison study is not find.
- Recommendations and future and prospective planning are necessary.
Response: Recommendations and future prospective have been added according to the suggestion of the reviewer
- Figures 1 and 3 express the same concept and may be combined.
Response: Action taken according to the suggestion of the reviewer.
- Figs. 2 and 3: The resolution and quality have been improved.
Response: We are thankful to the respectable reviewer for valuable comment.
- Add the type of solvent used in Table 1.
Response: Action taken according to the suggestion of the reviewer.
- In Tables 1, 2, and 3, replace "Authors" with "References"
Response: Replaced "Authors" with "References"
- Table 1: "Phytochemicals" is ambiguous; please clarify
Response: Changes have been made.
- Line 51: Replace ". [1]" with "[1].", then repeated for all references.
Response: Replaced ". [1]" with "[1]."
- Line 412-415” Keeping in mind the gap between protein………. accompanied by phytochemicals.” Move to the conclusion section.
Response: Action taken according to the suggestion of the reviewer.
Comments and Suggestions for Authors
Dear authors, this review manuscript contains relevant information, however, I believe that the following aspects should be reviewed and modified.
The authors are highly thankful to the respected reviewer for their kind suggestions/comments for improving the manuscript.
1.- It is recommended to improve the image of figure 2, because the chemical structures are not visible.
Response: We appreciate the reviewer for the relevant information. The figure 2 has been improved according to suggestion of reviewer.
2.- The Bioactive components in different vegetables sprouts section it would be convenient, it is indicated that they have a large number of bioactive compounds, can information on the concentrations be included?
Response: Yes! We added the information in detail about the different Bioactive components in different vegetables sprouts. But we did not add numeric value of different bioactive compounds.
3.- In line 261 is convenient to modify bioavailability by Bioavailability . Thus, in this section it would be convenient include more information on other compounds (as poliphenolic compounds, among others) in addition to broccoli iso-thiocyanate, or if this a compound found in all sprouts, include the corresponding citations.
Response: The authors are highly thankful to the respected reviewer for their kind suggestion. Improvements have been made in the revised manuscript.
4.- In section 5.1. Bioavailability of sprout against Brain issues, only anthocyanins have this effect? ​​There is no other associated antioxidant or polyphenolic group.
Response: In section 5.1. Different components have been added.
5.- In the Compensatory Sprouting as a Potential Therapeutic Strategy for Amyotrophic Lateral Sclero section, it would be convenient include compounds present in the sprouts that generate this effect.
Response: We thankful the reviewer for the valuable comments. Data has been added.
6.- It would be appropriate include the full word of GIT, before only including the acronym.
Response: Changes have been made according to suggestion of reviewer.
7.- In section 5.3. Bioavailability of sprout against GIT health problems, the effect is mainly related to fiber content, however, the generation of short-chain fatty acids as a result of their metabolism is not highlighted, as well as the modification of spices bacterial from microbiota that generates a beneficial effect in the prevention of GIT diseases.
Response: We thankful the reviewer for the valuable comments. Data about the role of fiber in GIT has been added in the revised manuscript.
On the other hand, in this same section, it would be convenient to include information on polyphenolic compounds that can generate a beneficial effect.
Response: We thankful the reviewer for the valuable comments. Data about the role of polyphenolic compounds in GIT has been added in the revised manuscript.
8.- In the Bioavailability of sprout against CVDs section, it would be convenient to include the full word CVDs, followed by the acronym.
Response: We appreciate the reviewer for the valuable information. Changes have been made according to suggestion of reviewer.
9.- Review the wording of line 343 ; Broccoli and broccoli sprouts contain an antioxidant that….
In this same sentence, you could indicate which antioxidant compounds are attributed to the prevention of heart disease, hypertension, and stroke.
Response: The authors are highly thankful to the respected reviewer for their kind suggestion. Changes have been made according to suggestion of reviewer.
10.- Modify in table 3 all in vitro by cursive format (in vitro)
Response: We appreciate the reviewer for the relevant information. Modified from in vitro to in vitro

Reviewer 2 Report
Dear authors, this review manuscript contains relevant information, however, I believe that the following aspects should be reviewed and modified.
1.- It is recommended to improve the image of figure 2, because the chemical structures are not visible.
2.- The Bioactive components in different vegetables sprouts section it would be convenient, it is indicated that they have a large number of bioactive compounds, can information on the concentrations be included?
3.- In line 261 is convenient to modify bioavailability by Bioavailability . Thus, in this section it would be convenient include more information on other compounds (as poliphenolic compounds, among others) in addition to broccoli iso-thiocyanate, or if this a compound found in all sprouts, include the corresponding citations.
4.- In section 5.1. Bioavailability of sprout against Brain issues, only anthocyanins have this effect? ​​There is no other associated antioxidant or polyphenolic group.
5.- In the Compensatory Sprouting as a Potential Therapeutic Strategy for Amyotrophic Lateral Sclero section, it would be convenient include compounds present in the sprouts that generate this effect.
6.- It would be appropriate include the full word of GIT, before only including the acronym.
7.- In section 5.3. Bioavailability of sprout against GIT health problems, the effect is mainly related to fiber content, however, the generation of short-chain fatty acids as a result of their metabolism is not highlighted, as well as the modification of spices bacterial from microbiota that generates a beneficial effect in the prevention of GIT diseases.
On the other hand, in this same section, it would be convenient to include information on polyphenolic compounds that can generate a beneficial effect.
8.- In the Bioavailability of sprout against CVDs section, it would be convenient to include the full word CVDs, followed by the acronym.
9.- Review the wording of line 343 ; Broccoli and broccoli sprouts contain an antioxidant that….
In this same sentence, you could indicate which antioxidant compounds are attributed to the prevention of heart disease, hypertension, and stroke.
10.- Modify in table 3 all in vitro by cursive format (in vitro)
Author Response

(The authors gave the same response as above.)

Round 2
Reviewer 1 Report
- "novel extraction" Changes through manuscript to "new"
- Figure 1: The chemical structure's clarity is weak.
Author Response
Comments and Suggestions for Authors
- "novel extraction" Changes through manuscript to "new"
Response: Changes have been made according to suggestion of reviewer.
- Figure 1: The chemical structure's clarity is weak.
Response: Clarity has been made according to suggestion of reviewr.
Comments and Suggestions for Authors
It is necessary to modify the word in vitro in italic format in different sections of the manuscript.
Response; The word (in vitro) has been modified in different sections of the manuscript

Reviewer 2 Report
It is necessary to modify the word in vitro in italic format in different sections of the manuscript.
Author Response

(The authors gave the same response as above.)
